# Role of circRNA in E3 Modification under Human Disease

**DOI:** 10.3390/biom12091320

**Published:** 2022-09-18

**Authors:** Zishuo Chen, Minkai Song, Ting Wang, Jiawen Gao, Fei Lin, Hui Dai, Chao Zhang

**Affiliations:** 1Guangdong Provincial Key Laboratory of Single Cell Technology and Application, Southern Medical University, Guangzhou 510515, China; 2Division of Orthopaedic Surgery, Department of Orthopaedics, Nanfang Hospital, Southern Medical University, Guangzhou 510515, China; 3Division of Spinal Surgery, Department of Orthopaedics, Nanfang Hospital, Southern Medical University, Guangzhou 510515, China; 4Hospital Office, Ganzhou People’s Hospital, Ganzhou 341000, China; 5Hospital Office, Ganzhou Hospital-Nanfang Hospital, Southern Medical University, Ganzhou 341000, China

**Keywords:** circRNA, ubiquitination machinery, cell signal transduction, ubiquitin system, miRNA

## Abstract

Circular RNA (circRNA) is often regarded as a special kind of non-coding RNA, involved in the regulation mechanism of various diseases, such as tumors, neurological diseases, and inflammation. In a broad spectrum of biological processes, the modification of the 76-amino acid ubiquitin protein generates a large number of signals with different cellular results. Each modification may change the result of signal transduction and participate in the occurrence and development of diseases. Studies have found that circRNA-mediated ubiquitination plays an important role in a variety of diseases. This review first introduces the characteristics of circRNA and ubiquitination and summarizes the mechanism of circRNA in the regulation of ubiquitination in various diseases. It is hoped that the emergence of circRNA-mediated ubiquitination can broaden the diagnosis and prognosis of the disease.

## 1. Introduction

### 1.1. circRNA Participates in Multiple Functions of Cellular Physiological Regulation

circRNA is a kind of long, closed noncoding RNA in which the 5′ and 3′ ends are covalently linked by the back-splicing of exons from a single pre-mRNA [1,2]. circRNA is widely expressed in mammalian cells and participates in the regulation of various biological processes. Compared with linear RNA, circRNA lacks the 5′cap and 3′tail and is therefore more resistant to RNase degradation [3]. circRNA has multiple functions in cells and physiology, most notably acting as a “sponge” and inhibiting the activity of one or more miRNAs [4,5,6,7]. Hansen et al. first discovered that circRNA acts as miRNA sponges in CDR1as [8,9,10]. In the cytoplasm, circRNA functions as competing endogenous RNAs that bind miRNAs, preventing them from binding or inhibiting their target mRNAs, thereby affecting their transcription or translation [11,12]. Conversely, circRNA can also affect their function by sequestering proteins [13]. For example, the cytoplasmic localized circPABPN1 results in reduced mRNA translation by inhibiting the binding of RBP HuR (also known as ELAVL1) to its cognate linear PABPN1 mRNA [14]. In addition, circRNA can also bind to different proteins by acting as protein scaffolds, such as circAmotl1 binds to the kinases AKT1 and phosphoinositide-dependent kinase 1 (PDK1), facilitating the cardio-protective nuclear translocation of pAKT [15]. In tumor immunity, circRNA regulates proliferation through its effects on signaling pathways, transcription factors, and cell cycle checkpoint regulators, and is involved in tumorigenesis and progression [16]. In addition, circRNA inhibits the cell cycle by promoting the interaction of binding proteins with ubiquitin ligases [17], cell repair, and regeneration [18]. circRNA also participates in the activation of macrophages through ubiquitination modification [19,20,21], thereby promoting the proliferation and migration of fibroblasts, and can be used as targeted anti-inflammation therapy. Studies have also shown that the circRNAs regulating protein ubiquitination and degradation and cellular immune responses are involved in the regulation of a variety of tumor immunity [22,23].

### 1.2. The Process of Ubiquitination Modification

In a multi-step process, ubiquitin is covalently linked to the lysine residues of the substrate protein. If a single ubiquitin molecule is connected to a protein, it is called monoubiquitination, and the lysine residue of the ubiquitin molecule attached to the substrate can act as a receptor for another ubiquitin molecule, which can be repeated to form polyubiquitinated proteins [24]. The poly ubiquitin chain serves as a recognition signal of the 26S proteasome. The 26S proteasome is the main regulator of protein abundance in the cell, so poly ubiquitination usually initiates the proteolysis of the substrate [25]. The human genome encodes an estimated 500–600 ubiquitin ligases, which are equivalent to the predicted 518 kinases. Ubiquitination is a highly dynamic process that is equilibrated by the uncoupling of ubiquitin by deubiquitinating enzyme (DUB). It is estimated that more than 100 DUBs, which fall into seven subgroups: the ubiquitin-specific proteases (USPs), the ubiquitin C-terminal hydrolases (UCHs), the ovarian tumor proteases (OTUs), the Machado-Josephin domain proteases (MJDs), the JAB1/MPN+/MOV34 (JAMM) domain proteases, the monocyte chemotactic protein-induced proteins (MCPIPs), and the motif interacting with ubiquitin-containing DUB family (MINDY) [26], are responsible for the reversible nature of ubiquitin modification, and play important roles in recovering ubiquitin from the proteasome substrate, stabilizing protein by counteracting its polyubiquitination, and preventing proteolysis-ubiquitous [27,28]. DUB constitutes the ubiquitin-proteasome system with E1, E2, and E3 enzymes, which together regulate ubiquitination modification [29].

Two E1 ubiquitin activating enzymes, 38 E2 conjugating enzymes and 600–1000 E3 ubiquitin ligases, have been found in humans [30]. Humans have two types of E1: UBA1 (also known as UBE1) and UBA6 (also known as UBE1L2), which initiate the ubiquitination process in an ATP-dependent manner. E1 is highly conserved in humans, which initiates the ubiquitination process by activating ubiquitin. First, E1 binds ATP, and then the AMP moiety of ATP is ligated to ubiquitin’s carboxyl terminus, forming a ubiquitin adenylate that remains noncovalently bound to E1 and releasing pyrophosphate. Next, ubiquitin is transferred to the active-site cysteine residue, exchanging the high-energy acyl phosphoanhydride bond with AMP for a thioester bond, then AMP leaves the enzyme. Third, while one ubiquitin remains thioester-linked to E1, another ubiquitin adenylate is formed, filling the ubiquitin adenylate binding site left vacant after step two. This third step is not intrinsically necessary for ubiquitin’s activation, but under physiological conditions E1 exists primarily as a ternary complex of ATP, ubiquitin, and E1~ubiquitin [25,31]. A potent small-molecule inhibitor of E1 enzymes, TAK-243 has been reported to cause depletion of cellular ubiquitin conjugates, resulting in the disruption of signaling events, the induction of proteotoxic stress, cell cycle progression and DNA damage repair pathways [32]. Xu et al. reported that knocking down E1 can lead to the apoptosis of leukemia and multiple myeloma cells [31]. Therefore, The E1 enzyme is considered to be a new target for the treatment of leukemia and multiple myeloma.

UBE2B (also known as hHR6B or Rad6B) is an E2 enzyme which may play a role in breast cancer development. In addition, UBE2B also forms a complex with the ligase MDM2 to mediate the degradation of p53. E2 enzymes UBE2D2 (also known as UbcH5b) and UBE2D3 also cooperate with the ligase β-TrCP to degrade inhibitors [33,34]. The overexpression of UBE2D may lead to early breast cancer, in which they interact with MDM2 to mediate p53 ubiquitination and degradation. Human E2 enzymes UBE2C (also known as UbcH10) and UBE2S (also known as E2-EPF) and E3 ligase later promote the Anaphase-Promoting Complex/Cyclosome (APC/C) to coordinate the cell cycle process [30]. UBE2S elongates Ub chains on substrates and increases the rate of substrate ubiquitination [35,36]. In recent years, studies have reported that UBE2C is overexpressed in lung cancer, bladder cancer, and ovarian cancer [37]. Therefore, these enzymes may serve as potential biomarkers for clinical diagnosis.

E3s is a large class of enzymes which are specific to the target protein. This specificity distinguishes their functions in the ubiquitin-proteasome system (UPS) and stably regulates protein homeostasis. Due to their structural domains and biochemical characteristics, E3s are divided into three subfamilies, E6-AP C-terminal homology (HECT), a really interesting new gene (RING) and U-box [38]. HECT type E3 is found in humans with only 28 members, HECT-type E3s contain an active cysteine residue that forms a thioester bond with ubiquitin before the substrate interacts. The E3 ligase containing the RING domain is composed of more than 600 members, accounting for 95% of the E3 ligase. The Cullin RING ligase (CRL) is the largest RING-type E3 ligase family. RING-type E3s and U-box E3s make the E2-ubiquitin complex into close contact with the substrate to allow direct transfer of ubiquitin. U-box E3 is also regarded as a subtype of RING-type E3 ligase. Therefore, E3 ligase is considered as a molecular mediator and catalyst. The dual roles combine the correct E2 with the correct substrate resulting in the improvement of the transfer rate of ubiquitin. The abnormal regulation of E3 ligase has been shown to be related to the development of cancer, so it may become a potential target for cancer treatment [39,40,41,42].

DUB is an important part of the ubiquitin proteasome system (UPS) [43]. It can selectively cleave the isopeptide bonds presented at the C-terminus of ubiquitin molecules, then remove ubiquitin or ubiquitin-like proteins from various protein substrates, and regulate ubiquitination degradation to facilitate the expression and activity of many key molecular targets. The deubiquitinating enzyme USP18, like other ubiquitin-specific proteases, carries DUB activity. USP18 expression is up-regulated and increases the stability of a key ISG15-binding protein that promotes carcinogenesis to promote tumorigenesis [44]. In addition, DUBs are involved in the control of carcinogenesis. For example, USP5 negatively regulates the tumor suppressor p53, USP28 stabilizes myc, USP9X promotes β-catenin oncogenic signaling, etc. [45]. Therefore, DUB regulates a variety of cellular processes and functions, such as regulating gene expression, apoptosis, cell cycle, DNA repair, and cytokine signaling. Producing/releasing free Ubiquitin, the cleavage of polyubiquitinated chains and the complete removal of ubiquitin chains from ubiquitinated proteins are three different mechanisms of action that specifically remove ubiquitin from the substrate [46]. Due to the activity of DUB, they become attractive therapeutic targets for cancer and other diseases. Tumor suppressor protein targeting DUB may be a good activation target. DUB may play a role in inhibiting cancer, and inhibiting DUBs as oncoproteins or inhibiting their activation could be a promising therapeutic strategy. As a pan-DUB inhibitor, PR-619 effectively induces cytotoxicity and apoptosis in chemoresistant urothelial carcinoma (UC) cells in vitro and inhibits the growth of chemoresistant UC cells in vivo [47]. Pimozide, a specific USP1 inhibitor, blocks glioma stem cell maintenance and radio resistance [48], However, as nonspecific DUB inhibitors, these broad-spectrum inhibitors may amplify their biological effects and non-specific toxicity, so it is still recommended to use specific DUB inhibitors in clinical practice. Currently, many new screening methods have been developed and used to select small molecule inhibitors and compounds for DUB. High-throughput screening was used to identify small molecule inhibitors that selectively target Ub C-terminal hydrolase (UCH-L1). In addition, cell-based screens were also used to select compounds that induce cathepsin-dependent apoptosis. Although great progress has been made in the development of small-molecule inhibitors targeting DUBs, there is still a long way to go before clinical application [46,49]. In summary, ubiquitination is essential to maintain body homeostasis by controlling a wide range of cellular functions, and the ubiquitin system has received attention as a promising drug target [27].

## 2. Ubiquitination Is Essential for the Pathophysiological Regulation of Multiple Systems

Ubiquitination refers to the process by which ubiquitin molecules classify the proteins in the cell under the action of a series of special enzymes, select target protein molecules from them, and specifically modify the target protein [50,51,52]. Ubiquitin is an essential protein of 76 amino acids, and is highly conserved in its amino acid sequence [53]. Ubiquitination is essential for a variety of physiological processes, including cell survival and differentiation, and innate and adaptive immunity [54,55]. Moreover, the ubiquitin pathway is involved in the regulation of many basic cellular processes [56,57], including cell division and differentiation, immune response, autophagy, DNA repair, and apoptosis [48]. The processes of ubiquitination are separated into three basic steps, catalyzed by ubiquitin activating enzyme (E1), ubiquitin conjugating enzyme (E2) and ubiquitin ligase (E3) [58,59]. Polyubiquitination refers to the attachment of more than two Ub molecules to the same lysine residue of the substrate in a chain. Compared to monoubiquitination, there are many types of polyubiquitination, which can be linked by many lysine residues in Ub or through its N-terminal Met. To date, there are eight types of polyubiquitin linkages that have been identified (K6, K11, K27, K29, K33, K48, K63 and Met1) with specific functions.

Ubiquitin has a complex surface and contains seven lysine residues (Lys6, Lys11, Lys27, Lys29, Lys33, Lys48, and Lys63) or through the ubiquitin amino terminal Met1 residue (which generates linear chains), together with its amino terminus, provides eight attachment sites for further ubiquitin molecules [57]. Proteins can be ubiquitinated on single or multiple lysines; in addition to the homotypic chain produced by one lysine linkage of Ub, branched ubiquitination can be produced on two lysine residues of the same ubiquitin by multiple ligase actions. Ubiquitination of different structures can lead to different functions [60]. At the same time, the counter-balance to the process of ubiquitination is achieved by enzymes known as deubiquitinases (DUBs). In cells, deubiquitination enzymes work by removing ubiquitin tags from tagged proteins or reducing the length of polyubiquitin tags [49]. The balance between ubiquitination and deubiquitination affects physiological protein abundance and activity [61]. The critical role of the Lys48 and Lys63 linker chains in cell signaling has been well documented in many studies; in contrast to “atypical” chain types such as Lys6, Lys11, Lys27, Lys29, Lys33 or Met1, there are few reports, but a few studies have found that the physiological role of these atypical chains is also obvious [62,63,64,65]. Simultaneously, the specific role of ubiquitination in T cells, macrophages and DC cells is still unclear, as the physiological functions of multiple genes involved in ubiquitination in these immune cells have not been fully elucidated, especially for immunotherapy against ubiquitination, which requires further development [66]. Therefore, research on regulating ubiquitination can broaden the understanding of many diseases, such as the pathophysiological process of tumor immunity, inflammation, or those involving the nervous system [67].

## 3. circRNA Regulated Ubiquitination Modification

Accumulating research has revealed that circRNA regulated protein ubiquitination and degradation as well as participated in the process of tumor immunity. For example, Li et al. found that circNDUFB2 was down-regulated in non-small cell lung cancer and suggested a poor prognosis for patients. In vitro studies found that the overexpression of circNDUFB2 significantly inhibited the proliferation, migration and invasion of NSCLC cells and recognized circNDUFB2 as a suppressor in the progression of NSCLC. In order to explore the mechanism of circNDUFB2 in non-small cell lung cancer, the authors found that circNDUFB2 enhanced the ubiquitination and degradation of IGF2BPs. TRIM25 is a RNA-binding protein and belongs to the Tripartite Motif (TRIM) family of E3 ubiquitin ligases, which catalyzes the addition of polyubiquitin chains to its substrates for degradation. The results suggest that TRIM25 interacts with IGF2BP proteins and degrades them via the ubiquitin–proteasome pathway in NSCLC cells. The binding of circNDUFB2 to TRIM25 is dependent on its RNA-binding activity. CircNDUFB2 affects NSCLC progression by functioning as a scaffold to enhance the interaction between TRIM25 and IGF2BPs [68]. Another study has found that the expression of circFoxo3 is significantly up-regulated in tumor cell apoptosis. In order to explore the mechanism, the authors have found that the expression of circFoxo3 increases the expression level of Foxo3 protein, but inhibits the expression of p53 [69]. Among them, MDM2-induced ubiquitination and degradation play an important role. Oncoprotein MDM2 is the main E3 ubiquitin ligase of p53 tumor suppressor, and circFoxo3 promotes MDM2-induced p53 ubiquitination and subsequent degradation, resulting in a decrease in p53 expression, and circFoxo3 could prevent MDM2-induced Foxo3 ubiquitination and degradation, resulting in increased Foxo3 protein levels, which in turn induced apoptosis [70,71]. In addition, MDM2, the major E3 ubiquitin ligase of p53, is a key regulator of p53 stability via the 26S proteasomal degradation pathway and is overexpressed in a variety of tumors. MDM2 binds to p53 to regulate its transcriptional activity. Lou et al. found that circRNA CDR1as was able to disrupt the p53/MDM2 complex by interacting with the DBD domain of p53, thereby increasing p53 stability by inhibiting its MDM2-mediated ubiquitination and subsequent degradation. Therefore, CDR1as deletion may significantly promote tumorigenesis in gliomas through p53 inactivation. As can be seen, CDR1as plays a crucial role as a tumor suppressor in glioma tumorigenesis. Recently, it was found that circFoxo3 has a function whereby it binds to proteins in related signal pathways. For example, circfoxo3 leads to decreased FAK/PI3K/AKT pathway activity and increased aging-related nuclear morphology through interaction with the anti-aging proteins ID-1 and E2F1 and anti-stress proteins FAK and HIF1α. Furthermore, circfoxo3 ectopically expressed in the cytoplasm leads to the arrest of ID1, E2F1, HIF1α and FAK in the cytoplasm, thereby reducing the expression levels of these proteins in the nucleus, thus blocking the anti-aging function of these proteins and promoting cellular senescence. In addition, circRNA also makes a great contribution to hepatocellular carcinoma (HCC) malignant tumors. Sun et al. found that the high expression of circADD3 was negatively correlated with vascular infiltration, intrahepatic metastasis and distant metastasis through circRNA microarray expression profiles [72]. The exogenous expression of circADD3 significantly weakens the invasion and metastasis of HCC cells in vitro and in vivo. Histone methyltransferase EZH2 is an important oncoprotein. There is a strong negative correlation between the expression of EZH2 and circADD3. CircADD3 destabilizes EZH2 by enhancing CDK1-mediated ubiquitination of EZH2 and promotes the expression of several anti-metastatic genes in HCC through EZH2. The authors found that the ubiquitination level of EZH2 was significantly increased after the overexpression of circADD3, indicating that circADD3 affects the stability of EZH2 protein through ubiquitination. CDK1, a cyclin-dependent kinase, is an important regulator for EZH2 ubiquitination and subsequent proteasomal degradation via phosphorylation at Thr-345 and Thr-487 sites of EZH2.The authors demonstrated that CDK1 is required for circADD3-mediated ubiquitination and the degradation of EZH2. In addition, this study pointed out that circRNA directly binds to EZH2 and inhibits its protein expression, supporting the notion that circRNA functions not only by acting as miRNA sponges. Therefore, circADD3 has been identified as an effective biomarker for the diagnosis and prognosis of HCC.

circRNA mediated ubiquitination also plays an important role in other diseases. Zhou et al. found that circHECTD1 (HECT domain E3 ubiquitin) and HECTD1 participate in SiO_2_-induced macrophage activation through ubiquitination; the host gene of circHECTD1, HECTD1, is an E3 ubiquitin ligase that regulates cell migration mechanisms. After induction by SiO_2_, it was found that the expression of circHECTD1 was decreased in RAW264.7 cells, while the expression of HECTD1 was up-regulated, and the authors determined that HECTD1 was involved in circHECTD1-mediated macrophage activation, regulating the activation of RAW264.7 in response to SiO_2_ exposure. HECTD1 is involved in ubiquitination because it encodes a novel protein homologous to the E3 ubiquitin ligase contained in the E6-AP C-terminal (HECT) domain. ZC3H12A is a novel DUB whose activity is abolished by mutation of the CCCH zinc finger domain, and the authors found that HECTD1 mediates macrophage activation through ZC3H12A ubiquitination in response to SiO_2_ through experiments. SiO_2_-activated macrophages promote fibroblast proliferation and migration through the circHECTD1/HECTD1 pathway, elucidating the link between the activation by SiO_2_-macrophages and the circHECTD1/HECTD1 pathway [73]. It was also found that circRNA was highly expressed in the central nervous system and is involved in the regulation of certain pathophysiology. Zhang et al. found that circDYM is down-regulated in chronic unpredictable stress (CUS). As an endogenous miR-9 sponge, circDYM inhibits the activity of miR-9, which leads to an increase in the expression of E3 ubiquitin protein ligase 1 (HECTD1) in the downstream HECT domain, and an increase in HSP90 ubiquitination [74]. Furthermore, the authors found that LPS treatment of BV-2 cells decreased the ubiquitination of HSP90 and K63-linked Ub chains, and confirmed that HECTD1-ACT enhanced the interaction between HSP90 and K63-ubiquitin. In vivo experiments also showed that HSP90 expression was increased and K63 ubiquitination was decreased in the hippocampus of CUS mice compared with controls. CircDYM reduced glial cell activation, thereby participating in the activation of major depressive disorder (MDD). circRNA participates in protein ubiquitination and degradation by acting as an miRNA sponge, encoding proteins, etc., and plays an important role in various cellular regulatory activities such as tumor immunity and mediating inflammation. Therefore, circRNAs may be the key to disease diagnosis and treatment in terms of ubiquitination regulation. The mechanism of circRNA in ubiquitination needs to be studied further.

## 4. Mechanisms of circRNA Participation in Ubiquitination Modification

### 4.1. circRNA Acts as a microRNA Sponge for Ubiquitination

circRNA regulates ubiquitination in varied ways. First, circRNA acts as a miRNA sponge to participate in the process of ubiquitination. Huang’s research found that the loss of circNfix induced the differentiation and proliferation of cardiomyocytes [75]. The results suggested that the inhibitory effect of circNfix on cardiomyocyte proliferation may be through the enhancing of the ubiquitination of Ybx1. The authors found that the deletion of circNfix promotes cardiomyocyte proliferation in vitro, adult cardiomyocyte proliferation in vivo, and adult cardiac regeneration after myocardial infarction (MI), while overexpression of circNfix inhibits neonatal cardiomyocyte proliferation in vivo. To explore how circNfix exerts its function in vivo, the authors found that circNfix knockdown and overexpression could reduce and increase Ybx1 ubiquitination in cardiomyocytes. Moreover, Nedd4l was found to be one of the proteins identified by mass spectrometry analysis after circNfix RNA pulldown. Nedd4l is an E3 ubiquitin ligase that interacts with its substrate for ubiquitination. It was confirmed that the inhibitory effect of circNfix on cardiomyocyte proliferation was achieved by enhancing the ubiquitin-dependent degradation of Ybx1. E3 ubiquitin ligase 1 (HECTD1) in the HECT domain is a downstream target protein of miR-9. Zhang’s research suggested that circDYM binds to miR-9 and acts as an endogenous sponge to inhibit miR-9 activity, resulting in the increased ubiquitinated expression of HECTD1 and HSP90, thereby inhibiting microglial activation [74]. However, it is still unclear how the HECTD1 ubiquitination of HSP90 is involved in this process.

### 4.2. circRNAs Mediate Ubiquitination by Acting as Scaffolds for Proteins

Although most circRNA acts by acting as miRNA sponges, many studies have demonstrated different roles of circRNA than ceRNAs. circRNAs play important roles as a bridge in mediating ubiquitination by acting as a scaffold for proteins. Li et al. demonstrated that circNDUFB2 reduces IGF2BPs by forming a ternary complex of TRIM25/circNDUFB2/IGF2BPs [68]. The study found that circNDUFB2 was downregulated in NSCLC and is related to poor prognosis. Mechanistically, circNDUFB2 interacted with IGF2BPs, but was not regulated as an miRNA sponge. TRIM25 is the E3 ubiquitin ligase which interacts with IGF2BP proteins and degrades them via the ubiquitin–proteasome pathway in NSCLC cells. It was shown that circNDUFB2 acted as a scaffold to enhance the interaction between TRIM25 and IGF2BPs, which subsequently promoted TRIM25-mediated ubiquitination and degradation of IGF2BPs. Similarly, Shen et al. also found that circPDE4B acted as a scaffold to promote the binding of RIC8 guanine-nucleotide exchange factor A (RIC8A) and E3 ligase midline 1 MID1, thereby reducing the RIC8A-dependent activation of p38 mitogen-activated protein kinase signaling pathway and regulate OA progression [76]. CircPDE4B promotes the formation of a ternary complex between RIC8A and MID1. MID1 is the E3 ligase of RIC8A, and K415 is the main ubiquitination site of RIC8A, and the authors also found that circPDE4B overexpression or inhibition no longer regulated RIC8A after K415 mutation levels of ubiquitination, suggesting that circPDE4B acts as a scaffold to facilitate the association between RIC8A and MID1. Another study found that circRNA-DOPYE2 was significantly down-regulated in cisplatin-resistant esophageal squamous cell carcinoma (ESCC) cells and had a potential inhibitory role in mediating cisplatin resistance. Moreover, circRNA-DOPYE2 targets an oncogene, CPEB4, which can enhance cisplatin resistance in ESCC cells. Additionally, circ-DOPEY2 acted as a scaffold to enhance the interaction between TRIM25 and CPEB4, thereby potentiating TRIM25-dependent ubiquitination. Exploring its downstream mechanism demonstrated that circRNA-DOPYE2 increases cisplatin resistance in ESCC by inhibiting CPEB4-induced Mcl-1 translation [77]. Here we complement the potential mechanism by which circRNA functions as a scaffold for complexes, thereby promoting the ubiquitination of proteins, affecting the physiological functions of cells.

### 4.3. circRNAs Encode Proteins to Regulate Ubiquitination

circRNAs are generally considered to be noncoding RNAs. Studies on the translational relevance of circRNAs and the mechanisms involved in their encoding are limited. Studies found that a circRNA from the SNF2 histone linker PHD RING helicase (SHPRH) encodes a novel protein called SHPRH-146aa, and both circ-SHPRH and SHPRH-146aa are abundantly expressed in the normal human brain but downregulated in glioblastoma. The authors found that SHPRH-146aa protects full-length SHPRH from E3 ligase DTL-induced ubiquitination, thereby promoting PCNA turnover in vivo, inhibiting cell proliferation and tumorigenicity. The study showed that SHPRH-146aa could serve as a prognostic marker for glioblastoma [78]. In addition, the study of Zhang et al. identified a novel circRNA, circDIDO1, in gastric cancer, which was less expressed in gastric cancer, and this suggested a poor prognosis. Moreover, circDIDO1 encodes a novel tumor suppressor protein DIDO1-529aa, and RBX1 is an E3 ligase of the ubiquitination complex. The authors found that the DIDO1-529aa protein acted as a PARP1 inhibitor and promoted the ubiquitination and degradation of PRDX2 in GC cells. The PRDX2 downstream pathway inhibits GC cell growth and invasiveness [79]. In recent years, more and more studies have been conducted on cirRNAs, which can encode proteins. CircMAP3K4, which encodes the peptide circMAP3K4-455aa, was also found in hepatocellular carcinoma, and circMAP3K4-455aa promoted HCC growth in vitro and in vivo, in which circMAP3K4 translation is dependent on m6A modification and is mediated by the recruitment of IGF2BP1, which is an E3 ligase. The authors’ study found that MIB1 promotes circMAP3K4-455aa ubiquitination and degradation [19].

### 4.4. circRNA Is Able to Bind Directly to Proteins or mRNA to Influence Ubiquitination

circRNA can also bind directly to the ubiquitinated regions of proteins to participate in molecular regulation and affect downstream signaling pathways. The E3 ubiquitin ligase STUB1 has been shown to ubiquitinate HSP90, thereby targeting it to the proteasome for degradation. In the study of Xie et al., the authors found that circSHKBP1 regulates gastric cancer(GC) cell proliferation, migration and invasion in vivo and in vitro [80], and mechanistically, overexpression of circSHKBP1 reduced the amount of STUB1 binding to HSP90 by IP detection, verifying that circSHKBP1 and STUB1 are similar in the site competes with HSP90 for binding, leading to the conclusion that circSHKBP1 directly binds to HSP90 and inhibits the ubiquitination of HSP90 by STUB1, thereby accelerating GC development. AKT is one of the important effector elements of the canonical signaling pathway PI3K/AKT. The PI3K/AKT signaling pathway is considered to be a key pathway in tumor development and plays an important role in promoting tumor cell migration, invasion, survival and resistance, and AKT itself functions in an activated phosphorylated form. AKT has multiple ubiquitination sites; its K48-linked ubiquitination is targeted to the 26S proteasome, which directly leads to the degradation of the AKT protein, and the ubiquitination of K63 plays a key role in AKT signaling and protein transport. The study by Tang et al. found that hsa_circ_0124554 can specifically bind to AKT, reduced degradation of AKT by mediating site ubiquitination of AKT, and continued to continuously activate the downstream signaling pathways of AKT including GSK-3β and mTOR, thereby affecting the early metastasis of tumors [81]. Small ubiquitin-like modifier 2 (SUMO2) is a member of the SUMO family, which is a ubiquitin-like protein involved in post-translational protein modifications. SUMO can regulate a variety of cellular processes including transcription, maintenance of genome integrity and cell cycle by binding to a variety of proteins. SUMOylation is a ubiquitin-like protein modification. Glucose transporter proteins (GLUTs) are one of the most important transmembrane proteins in the human body, responsible for the transport and reabsorption of glucose in different tissues and organs of the body. GLUT1 is highly expressed in various tumors and is involved in regulating malignant phenotypes, such as tumor metastasis and proliferation. Moreover, circRNA can also bind to mRNA to promote protein ubiquitination. CircRNF13 is a circular splicing of exons 2–8 of the E3 ubiquitin protein ligase RNF13 gene. Mo et al. found that circRNF13 promotes the stability of SUMO2 by directly binding to the 3′-UTR of SUMO2 mRNA, increases SUMO2 protein expression, and promotes the ubiquitination of GLUT1 through SUMO2, which regulates the AMPK-mTOR pathway by inhibiting glycolysis, ultimately leading to NPC proliferation and metastasis [82].

## 5. The Effect of circRNA on Ubiquitination in Disease

### 5.1. circRNA Regulates the Effects of Ubiquitination on Cancer

The effect of circRNA on ubiquitination has some effect on the emergence and development of disease, particularly in cancer. The expression of circRNA-SORE was found to be up-regulated in sorafenib-resistant hepatocellular carcinoma. Mechanistically, it was found that circRNA-SORE functions by binding to YBX1, and both YBX1 and circRNA-SORE can be used as biomarkers for predicting sorafenib efficacy in HCC patients. The authors found that circRNA-SORE could stabilize YBX1 by preventing the degradation of YBX1 mediated by E3 ligase PRP19, and silencing circRNA-SORE in vivo could significantly improve the efficacy of sorafenib [83].

CircIL4R also plays an important role in the proliferation and metastasis of colorectal cancer (CRC). Colorectal cancer is one of the most common malignancies of the digestive system, and circIL4R is highly stable and upregulated in CRC. In CRC, CircIL4R acts as a sponge for miR-761, while the E3 ligase TRIM29 is a downstream target of miR-761, and circIL4R activates the PI3K/AKT signaling pathway through the TRIM29-mediated ubiquitination and degradation of PHLPP1, thereby promoting CRC cell proliferation and transfer in vivo [84].

Otubain-1 (OTUB1) is a deubiquitinating enzyme targeting NSCLC cells, and silencing OTUB1 completely abolished the PKP3 overexpression-mediated stabilization of the PD-L1 protein. Liu’s study found that circIGF2BP3 inhibited the anti-tumor immunity of NSCLC by increasing the expression of its targeted mRNA PKP3. Exploring its mechanism revealed that circIGF2BP3 could specifically sponge miR-328-3p and miR-3173-5p, alleviating their inhibition of PKP3, thereby inducing tumor immunosuppression in NSCLC. PKP3 stabilizes PD-L1 by promoting PD-L1 deubiquitination mediated by the deubiquitinase OTUB1, and PKP3 can also increase OTUB1 mRNA stability and upregulate OTUB1 expression, both of which synergistically induce tumor immunosuppression [85].

Pancreatic ductal adenocarcinoma (PDAC) is one of the leading causes of cancer death worldwide, and circRTN4 expression was found to be upregulated in PDAC and promote PDAC cell growth migration and invasion in vitro and PDAC tumor growth and liver metastasis in vivo. Exploring the mechanism found that circRTN4 acts as a sponge for miR-497-5p in PDAC cells, and there are studies that have found that miR-497-5p plays a tumor suppressor role in PDAC by targeting the oncogenic HOTTIP-HOXA13 pathway [86]. MiR-497-5p promotes the expression of lncRNA HOTTIP, RAB11FIP1 is a protein that interacts with circRTN4, ubiquitination of the lysine 578 (Lys578) residue of RAB11FIP1 is required for its degradation, and circRTN4 interacts with RAB11FIP1 and inhibits ubiquitination of RAB11FIP1 in PDAC to enhance its stability. Moreover, RAB11FIP1 promotes cancer migration and invasion by regulating the expression of the EMT-related proteins [87].

### 5.2. circRNA Regulates the Effects of Ubiquitination in Other Diseases

The regulation of ubiquitination by circRNA also has an effect in some other diseases, but not any that are as well-studied as cancer. In the regeneration of myocardial infarction, circNfix regulates the ubiquitination of Ybx1 in cardiomyocytes. While Ybx1 overexpression significantly increases cell proliferation, Ybx1 silencing blocks the circNfix knockout-induced proliferation of cardiomyocytes, and circNfix indirectly affects cardiomyocyte regeneration in this way [75]. The expression of circNfix significantly affected the interaction between Ybx1 and E3 ligase Nedd4l, and the inhibitory effect of circNfix on cardiomyocyte proliferation may be achieved by enhancing the ubiquitin-dependent degradation of Ybx1. In major depressive disorder, circRNA also has an impact on the development of the disease. CircDYM acts as an endogenous microRNA-9 (miR-9) sponge, inhibiting miR-9 activity, resulting in the increased expression of the downstream target HECT domain E3 ubiquitin-protein ligase 1 (HECTD1) and increased HSP90 ubiquitination, thereby reducing microglia activation (Table 1 and Figure 1).

As shown in Figure 1, most circRNAs such as circNDUFB2, circADD3, circNifx, etc. combine with ubiquitin ligase E3, such as TRIM25, to form complexes to regulate cell proliferation or apoptosis, thereby participating in the development of various diseases. Due to the specificity of the E3 ubiquitin ligase, most studies have found that circRNAs affect cell activities by directly or indirectly regulating E3 ligases. In the process of ubiquitination, E1, E2 enzymes and DUB are less studied than E3 [88]. Therefore, it is possible to explore whether non-coding RNAs are involved in other processes of ubiquitination modification.

## 6. Discussion

circRNA is a type of endogenous ncRNA with abundant and stable expression. With the development of sequencing technology, a variety of circRNA in the body can be easily identified. At present, the most researched is E3 ubiquitin ligase, which is currently considered to be a key regulator of cancer development [89,90,91]. It can enhance cell growth, accelerate cell cycle progress, promote migration and invasion, and inhibit cell apoptosis. Other members of ubiquitin ligases can also regulate protein stability, immune responses and cancer progression. The main research on the regulation of ubiquitination by circRNA is related to the E3 ubiquitin ligase, which recruits substrates and promotes or directly catalyzes the transfer of ubiquitin to targets. These enzymes largely determine the specificity of the ubiquitination reaction. Because E3s determine substrate specificity, their dysregulation by mutations, epigenetic changes, or transcriptional alterations may impair or accelerate the degradation of oncogenes and tumor suppressor genes, leading to disease. However, there are few studies on the regulation of ubiquitination by circRNA by binding E1, E2 and deubiquitination enzymes, and further research is needed to explore whether circRNA can participate in other aspects of ubiquitination regulation. Moreover, most of the current studies are aimed at circRNA as a miRNA sponge to regulate the expression of mRNA target genes. There are also studies that have shown that circRNA does not perform its function through ceRNA, but uses circRNA as a scaffold to regulate the ubiquitination of ubiquitin ligase-mediated proteins modification, and this mediating function of circRNA is essential. circRNA can not only participate in the regulation of ubiquitin ligase, but also participate in the ubiquitination modification of signal pathways in the occurrence and development of diseases, such as various types of tumors, immune inflammation, etc. However, although the role of circRNA in the development of ubiquitination and regulation of diseases is not yet clear, it is certain that circRNA can be used as a potential target for diagnosis or treatment. In summary, circRNA can act as a microRNA sponge to interact with proteins, and can also encode proteins or participate in transcriptional regulation and other ways to participate in the ubiquitination of proteins, thereby affecting the function of cells and playing an important role in a variety of diseases. Ubiquitination plays an important role in maintaining cellular homeostasis, including effects on cell signaling, apoptosis, protein processing, immune responses, and DNA repair. Therefore, it will lead to the occurrence of various diseases, such as affecting the occurrence and development of tumors, and the development of neurodegenerative diseases. Therefore, drug therapy targeting the ubiquitin-proteasome pathway offers potential opportunities for the treatment of tumors and neurodegenerative diseases. The emergence of circRNA has broadened the treatment of various diseases, such as cardiovascular and cerebrovascular diseases, neurological diseases, and various tumors. Although circRNA has been explored in various ways to regulate the ubiquitination process, their biological functions remain unknown. With the discovery of ubiquitination-related circRNA functions, many circRNAs have emerged as promising biomarkers for disease diagnosis and prognosis, as well as potential therapeutic targets.

## Figures and Tables

**Figure 1 biomolecules-12-01320-f001:**
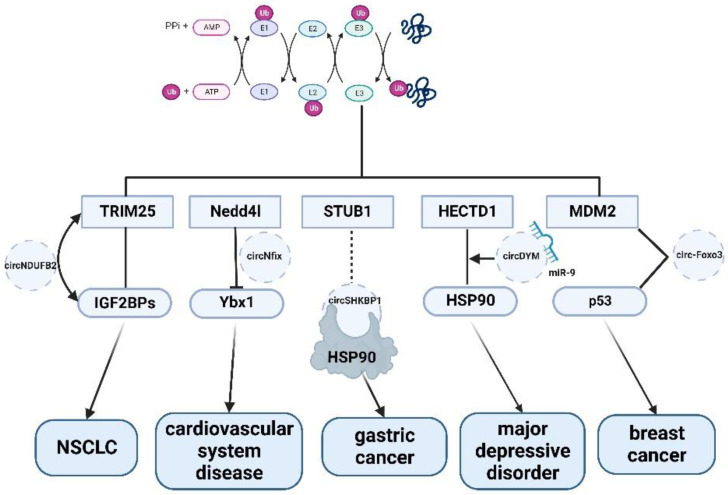
The mechanism of circRNA in ubiquitin signaling pathways.

**Table 1 biomolecules-12-01320-t001:** Summary of the regulatory mechanism based on circRNA involved in ubiquitination modification.

circRNA Involved in Ubiquitination	Disease	Regulation Mechanism	References
circNDUFB2	non-small cell lung cancer (NSCLC)	Enhanced ubiquitination modification of tumor regulator IGF2BPs	circNDUFB2 inhibits non-small cell lung cancer progression via destabilizing IGF2BPs and activating anti-tumor immunity
circSHKBP1	gastric cancer	Inhibits the ubiquitination of HSP90	Exosomal circSHKBP1 promotes gastric cancer progression via regulating the miR-582-3p/HUR/VEGF axis and suppressing HSP90 degradation
circ-FOXO3	breast cancer	Promotes p53 ubiquitination induced by E3 enzyme MDM2	Induction of tumor apoptosis through a circular RNA enhancing Foxo3 activity
circADD3	hepatocellular carcinoma	Enhance CDK1-mediated EZH2 ubiquitination and promote the expression of anti-metastatic genes	Circular RNA circ-ADD3 inhibits hepatocellular carcinoma metastasis through facilitating EZH2 degradation via CDK1-mediated ubiquitination
circHECTD1	silicosis	CircHECTD1 mediates HECTD1 and activates macrophages through ZC3H12A-mediated HECTD1 ubiquitination in response to SiO2 exposure	circRNA Mediates Silica-Induced Macrophage Activation Via HECTD1/ZC3H12A-Dependent Ubiquitination
circDYM	major depressive disorder	Inhibit the activity of miR-9 and increase HSP90 ubiquitination	CircDYM ameliorates depressive-like behavior by targeting miR-9 to regulate microglial activation via HSP90 ubiquitination
circNfix	cardiovascular system	circNfix enhances the interaction of Ybx1 with Nedd4l and induces Ybx1 degradation via ubiquitination and inhibits the expression of cyclin A2 and cyclin B1	Loss of uper-enhancer-egulated circRNA Nfix induces cardiac regeneration after myocardial infarction in adult mice

## Data Availability

Not applicable.

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
