# Peer review of "Role of circRNA in E3 Modification under Human Disease"

_biomolecules, 2022, doi:10.3390/biom12091320_

Round 1

Author Response

Dear reviewers:

Thank you for giving us the opportunity to submit a revised draft of the manuscript “Role of circRNA in E3 modification under human disease” to Biomolecules. We appreciate the time and effort that you and the reviewers dedicated to providing feedback on our manuscript and are grateful for the insightful comments on and valuable improvements to our paper. We have incorporated most of the suggestions made by the reviewers. Those changes are highlighted within the manuscript. Please see below, in red, for a point-by-point response to the reviewers’ comments and concerns. All page numbers refer to the revised manuscript file with tracked changes.Please see the attachment.

Reviewer 2 Report

Summary:

 The review concerns the role of circRNA in regulating the ubiquitination of proteins and their function in biological processes involved in the progression of various diseases. This is a very good review of an interesting and important subject. The field is described in a clear and understandable fashion. Publication is recommended upon minor revision, as described below.

Broad comments:

The English should be edited. Parts 1 and 2 could be streamlined and partly combined. Sections 1.2 and 2 have redundancies and could be reorganized, condensed and partly combined. For instance, the basic ubiquitination reaction sequence (involving E1, E2, and E3) is outlined multiple times. One good description in the beginning would suffice.

Specific comments:

Line 76: “… E3 ubiquitination ligase …” should be “… E3 ubiquitin ligase ...”

Line 80: “… different ubiquitination genes …” should be “… different ubiquitin genes …”

Line 94: “eukaryote” should be “eukaryotes”

Line 96: “mono ubiquitination” should be monoubiquitination

Figure 1 is a bit misleading in that most E3 ubiquitin ligases (notably, RING domain E3 proteins) do not form a covalent linkage with ubiquitin but rather serve as adaptor proteins to recognize the substrate, binding both the substrate and the E2 enzyme, thus facilitating transfer of ubiquitin from the E2 enzyme to the substrate.

Author Response

Dear reviewers:

Thank you for giving us the opportunity to submit a revised draft of the manuscript “Role of circRNA in E3 modification under human disease” to Biomolecules. We are very much thankful to the reviewers for their deep and thorough review. I have revised my present review paper in the light of your useful suggestions and comments. I hope my revision has improved the paper to a level of your satisfaction. Please see the attachment.

Reviewer 3 Report

The authors review the function of circRNAs in the ubiquitination of target proteins. Disruption of this process results in numerous disease processes, which are described.  While the concept of the review is interesting and well-structured, the writing leaves much to be desired. Because of this deficiency, the concepts do not come through clearly and the reader is left to hopefully interpret the sentence and scientific facts correctly.  

Incomplete sentences:

Cleavage of fusion proteins to produce ubiquitinated molecules, as well as ubiquitin proteins with different conformations and proteins containing ubiquitin-like domains (UbLD), as well as the interaction of single ubiquitin or poly ubiquitin chains to regulate ubiquitination Retouch.”

“Ubiquitination modification in biological” – this is particularly problematic because it is a section heading.

Missing periods or problems with sentence structure:

Line 82, “specific Cleavage” “

Line 177, “E1 The”

Awkward sentence:

Line 201-5, "By binding to p53, MDM2 prevents the action of p53 and transports it from the nucleus to the cytoplasm, where MDM2 acts as an E3 ubiquitin link Enzyme and covalently link ubiquitin to p53, marking p53 to be degraded by the proteasome, resulting in an increase in Foxo3 protein level."

Incomplete or inaccurate: 

Cullin- RING ligase (CRL)

APC/C stands for Anaphase-Promoting Complex/Cyclosome

The statement “Proteasome ubiquitination” is mentioned several times. However, the proteasome is not being ubiquitinated in what the authors are described and not in the section titled “CircRNA is involved in the proteasome ubiquitination process as a microRNA sponge”. Perhaps the authors meant “proteasome-dependent degradation” or “ubiquitin-proteasome system”

This phrase is incorrect: “E2 catalyzes the transfer of ubiquitin from the E1 site to the active site on E2 through a transesterification reaction.” It is transthioesterification.

Minor comment:

The authors mention in Line 77,  “At this point, the ubiquitination process is completed.” It is only completed for monoubiquitination. This sentence doesn't provide the opening needed for polyubiquitination that is elaborated on below. 

Author Response

Dear reviewers:

Thank you for giving us the opportunity to submit a revised draft of the manuscript “Role of circRNA in E3 modification under human disease” to Biomolecules. We appreciate the time and effort that you and the reviewers dedicated to providing feedback on our manuscript. After receiving your reply email, we have carefully reviewed the manuscript and made revisions immediately, and supplemented relevant documents as required by the journal. Those changes are marked in red within the manuscript.Please see the attachment.

Reviewer 4 Report

The review written by Chen et al., has a lot of information which is up to date and beneficial to researchers studying circRNAs. However, the organization of the content is not good, the language and grammar is also difficult to follow in many places. A lot of ubiquitin related nomenclature like deubiquitination enzymes, E3 ubiquitinase etc is incorrect. Reorganisation of content and improvement of writing style and language is needed before publication.

Line 77: The last sentence of the paragraph is not required

Line 81: The sentence “At the same time, in cells, ……” is out of context; can be removed.

Line 92: The section on “ubiquitin regulation in biological systems” should precede the paragraphs on its role in pathophysiology (line 59). Also the title “ubiquitin regulation in biological ..” is incomplete.

Line 187: E3 ubiquitin ligase is wrongly written as ubiquitinase.

Section 3: circRNA regulated ubiquitination –should be divided into shorter paragraphs with subheadings-(1) circRNA NDUFB2 mediated regulation of IGF2BPs, (2) circRNA CDR1 mediated regulation of p53 etc where the authors should first mention which molecules are regulated by the circRNA and then its relevance in disease.

Line 221: The word activated is missing in SiO2-macrophages

Line 374: can be rewritten in better English

Line 350: Section 5: circRNA in diseases should be divided into paragraphs following figure 1: cancer, cardiovascular disease, lung cancer etc

Author Response

Dear reviewers:

Thank you for giving us the opportunity to submit a revised draft of the manuscript “Role of circRNA in E3 modification under human disease” to Biomolecules. We appreciate the time and effort that you and the reviewers dedicated to providing feedback on our manuscript and are grateful for the insightful comments on and valuable improvements to our paper. We have incorporated most of the suggestions made by the reviewers. Those changes are marked in red within the manuscript.Please see the attachment.

Round 2

Author Response

Dear reviewers:

Thank you for giving us the opportunity to submit a revised draft of the manuscript “Role of circRNA in E3 modification under human disease” to Biomolecules. We appreciate the time and effort that you and the reviewers dedicated to providing feedback on our manuscript. After receiving your reply email, we have carefully reviewed the manuscript and made revisions immediately, and supplemented relevant documents as required by the journal. Those changes are marked in green within the manuscript.Please see the attachment.

Reviewer 4 Report

The authors have made the necessary changes in the revised version. I have no further comments

Author Response

Dear reviewers:

Thank you for giving us the opportunity to submit a revised draft of the manuscript “Role of circRNA in E3 modification under human disease” to Biomolecules. We appreciate the time and effort that you and the reviewers dedicated to providing feedback on our manuscript.